



# Geomorphological activity and stability of surfaces and soils formed in hyperarid alluvial deposits (Atacama Desert, Chile)

Linda Andrea Elisabeth Maßon[1], Simon Matthias May[1*], Svenja Riedesel[1], Willem Marijn van der Meij[1], Stephan Opitz[1], Andreas Peffeköver[1,2], Tony Reimann[1]

[1]Institute of Geography, University of Cologne, Cologne, Germany

[2]Department of Earth Sciences, Simon Fraser University, Burnaby, Canada

*Correspondence to*: Simon Matthias May (mays@uni-koeln.de)

**Keywords:**

Hyperarid earth surface processes, feldspar single-grain luminescence, post-depositional mixing, Coastal Cordillera

**Abstract**

The hyperarid conditions of the central Atacama Desert, characterized by extremely low precipitation and high evaporation rates, create a unique environment where geomorphic stability is generally considered to be exceptionally high. Terrestrial cosmogenic nuclide-based surface exposure ages suggest that many surfaces underwent limited to no geomorphic changes

since the Neogene or early Pleistocene. However, a number of recent studies reveal more recent landscape-scale geomorphic activity and link this to slightly wetter episodes during the Quaternary. In order to determine drivers of geomorphic activity, we performed a multi-proxy analysis of five profiles situated in alluvial deposits along a climatic transect from the coastal plain to the upper reaches of the Coastal Cordillera (~0–2000 m a.s.l.), combining single-grain feldspar luminescence dating with geochemistry, granulometry, and field observations. Alluvial deposits are prone to heterogeneous bleaching; therefore,

we tested the bleachability of the feldspars and found a shallow effective bleaching depth (<2 cm) and high near-surface bleachability. The five profiles studied could be grouped into geomorphological active and stable sites, based on our multi-method approach. Profiles located on geomorphologically active surfaces show evidence of recent sediment deposition and, in one case, vertical grain transport. In contrast, stable surfaces preserve reworking signals relating to bioturbation at the coast and desert pavement formation in the hyperarid Coastal Cordillera. While no clear chronological trend along the west–east

climatic gradient could be found, two phases of widespread geomorphic surface activity – ~50 ka and during the last ~5 ka – coincide with regionally wetter intervals compiled from other studies. Our findings highlight the value of single-grain luminescence data for reconstructing local sediment dynamics and soil reworking in arid environments, and the need to account for both depositional and post-depositional processes in paleoenvironmental interpretations.



## 1 Introduction

The Atacama Desert in northern Chile is one of the driest regions on Earth, with an extremely hyperarid core receiving less than 2 mm of rainfall per year (Houston, 2006). Even though the timing of the onset of predominantly hyperarid conditions and their past variability remain highly debated (Ritter et al., 2019), these conditions have been more or less stable for millions of years (e.g. Dunai et al., 2005; Evenstar et al., 2017; Hartley and Chong, 2002; Nishiizumi et al., 2005; Rech et al., 2006). According to cosmogenic nuclide dating, the combination of hyperaridity and climatic stability is related to exceptionally low erosion rates (Jungers et al., 2013; Placzek et al., 2010), supporting the notion that virtual surface stability has persisted since the onset of hyperaridity (Nishiizumi et al., 2005). Nevertheless, wetter climatic episodes occurred during the Quaternary, related to the local activation and/or intensification of (water-driven) surface processes (e.g. Gayo et al., 2012; May et al., 2020; Medialdea et al., 2020; Nester et al., 2007; Pfeiffer et al., 2021, 2018; Ritter et al., 2019; Sáez et al., 2016). Climatic fluctuations during the Pleistocene and Holocene are documented in lake and wetland archives (Grosjean et al., 2001; Pfeiffer et al., 2018), rodent midden pollen records (Díaz et al., 2012; Maldonado et al., 2005), rodent midden fecal pallet size record (González-Pinilla et al., 2021), palaeosols (Veit, 1996), and fluvial and geoarchaeological evidence (Gayo et al., 2023, 2012; Latorre et al., 2013; Nester et al., 2007; Seeger et al., 2024; Vargas et al., 2006).

Most studies addressing fluvial and alluvial dynamics during the Late Pleistocene and Holocene focus on the Precordillera and the eastern desert margins, where Andean runoff exerts strong influence. More recently, Bartz et al. (2020a, 2020b), Haug et al. (2010) and Walk et al. (2020) have extended this work to coastal alluvial fan systems. Haug et al. (2010) showed the possibility of surfaces reactivation during El Niño events, whereas Walk et al. (2023) reported active soil processes that do not result in geomorphic changes of the landform. Yet, the timing and drivers of both depositional and post-depositional (i.e., soil) processes in the Coastal Cordillera remain poorly constrained, especially above the marine boundary layer, where fog and fog-depending loma vegetation are virtually absent (e.g. Diederich et al., 2020; Ritter et al., 2019; Vargas et al., 2006). To address this knowledge gap, we investigate soil and surface dynamics along a climatic transect spanning from the coastal plain to the hyperarid parts of the Coastal Cordillera. We focus on five profiles in alluvial deposits, combining stratigraphic observations, granulometry, geochemistry, and single-grain feldspar luminescence dating techniques.

In contrast to other dating methods, single-grain luminescence dating measures the burial time of individual sediment grains, which makes it a valuable tool for detecting both discrete depositional events and gradual reworking or soil mixing processes (e.g., Gliganic et al., 2016; Gray et al., 2019; Reimann et al., 2017; van der Meij et al., 2025). Although typically applied in more humid environments with typically more active geomorphic and pedogenic processes, this method also holds promise for the Atacama Desert (Zinelabedin et al., 2022, 2025). Therefore, the aims of this study are 1) to disentangle depositional and post-depositional sediment dynamics, and 2) assessing the applicability of single-grain luminescence dating as a sediment tracer and paleoclimatic proxy in a challenging mineralogical and climatic environment. Ultimately, this work contributes to





a better understanding of geomorphic surface stability, sediment mixing, and soil formation in arid and hyperarid landscapes, and their sensitivity to past climatic fluctuations.

## 2 Physical setting

### 2.1 Climate

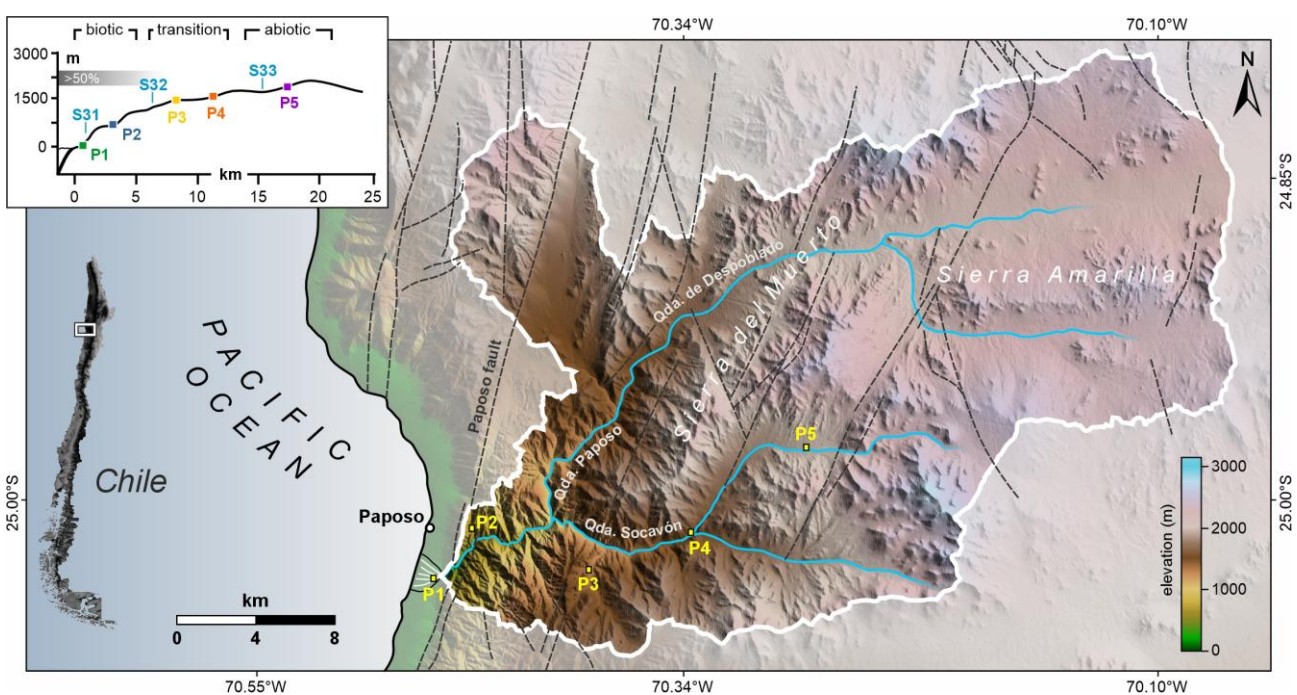

**Figure 1: Entire catchment of the Paposo Transect with the topography along the Paposo transect, including sampling and weather station locations as an inset. Fog occurrence as a grey bar in the inset (approx.) based on Cereceda et al. (2008). Biotic sampling sites P1 and P2 (fog occurrence & loma vegetation) are located below, transitional sampling sites P3 and P4 and hyperarid sampling site P5 above the marine boundary layer.**

The Atacama Desert is known for its hyperaridity, which results from subtropical atmospheric subsidence, coastal upwelling, the cold Humboldt current, and the Andean rain shadow (Garreaud et al., 2010; Houston, 2006; Houston and Hartley, 2003). While hyperaridity has established around or before the mid-Miocene (de Porras et al., 2017; Dunai et al., 2005; Rech et al., 2019), several studies documented climatic fluctuations on more recent time scales (e.g., Latorre et al., 2013, 2002; Nester et al., 2007; Pfeiffer et al., 2018; Vargas et al., 2006; Veit, 1996). Fluctuations in fog and rainfall are controlled by the complex interplay of Eastern Pacific Sea Surface Temperatures (SSTs), Pacific Decadal Oscillation (PDO) and El Niño Southern Oscillation (ENSO), South American Summer Monsoon (SASM) patterns as well as moist northerlies (MN) and additionally depend on season, altitude and latitude (e.g., Garreaud et al., 2008, 2010; Jara et al., 2020; Latorre et al., 2011; Vicencio Veloso et al., 2024). These climatic drivers control the moisture supply and consequently vegetation occurrence, which affects surface processes such as fluvial activity and pedogenesis along the transect.





On a local level, more humid areas can be found. In the Paposo transect, the Coastal Cordillera separates the narrow coastal plain from the broad Central Valley (Houston and Hartley, 2003). Marine fog frequently reaches elevations of up to ~1200 m a.s.l. (Cereceda et al., 2008; del Río et al., 2018), supporting the occurrence of loma vegetation and microbial activity in the lower part of the transect (González et al., 2011; Jaeschke et al., 2024; Merklinger et al., 2020; Pinto et al., 2006; Sun et al., 2024). This creates a strong moisture gradient along the transect, with average relative humidity values ranging from 76.1 %

at 160 m a.s.l. to 17.1 % at 1746 m a.s.l. (Table 1, Fig. S1).

**Table 1: Weather station data (Hoffmeister, 2018a, 2018b, 2018c). For location see Fig. 1. Min and mean air temperature as well as mean rel. humidity were recorded between 01. May 2021 and 01. September 2023.**

| Weather station number | Latitude | Longitude | Elevation a.s.l. (m) | Min. air temp | Mean air temp | Mean rel. humidity (air) |
|---|---|---|---|---|---|---|
| S31 | -25.1151° | -70.458° | 160 | 5.4 °C | 14.9 °C | 76.1 % |
| S32 | -25.101° | -70.401° | 1011 | 2.4 °C | 15.5 °C | 48.4 % |
| S33 | -25.0916° | -70.2786° | 1746 | -5.2 °C | 15.9 °C | 17.1 % |

## 2.2 Geology and geomorphology

Given the extensive drainage basin of the studied transect (~780 km²; Walk et al., 2023; Fig. 1), the lithological composition within the catchment is heterogeneous (Alvarez et al., 2016; Escribano et al., 2013). In the upper reaches, broad, shallow valleys are filled with semi-consolidated Miocene to Pliocene alluvial deposits, underlain by volcanic rocks of the La Negra Formation. In contrast, the central and lower parts of the catchment are primarily underlain by the Matancilla intrusive complex, composed predominantly of granodiorites and diorites of Middle to Upper Jurassic age. Additionally, Palaeozoic

metasedimentary rocks of the Chañaral Epimetamorphic Complex crop out in the north-western sector of the catchment (Alvarez A. et al., 2016).

The study area is crossed by the Paposo fault, which is trending NNE–SSW to NE–SW and being a part of the Atacama Fault Zone, a principal tectonic structure stretching from 20°30'S at Iquique to 29°45'S at north of La Serena (Alvarez et al., 2016; Escribano et al., 2013; Hervé, 1987; Scheuber and Andriessen, 1990). Several other minor faults cross the research area trending in the same direction (Alvarez et al., 2016; Escribano et al., 2013). Except for **P2**, all field sites within our study are

not directly intersected by active faults. The elevation profile of the transect is characterised by an abrupt transition from the coastal plain to the coastal cliff at about 110 m a.s.l.

## 3 Methods
### 3.1 Field work

Field work was conducted during September 2022 and March 2023 and included geomorphological survey, site documentation, and sampling. In total five soil or surface profiles were dug along a climatic and altitude transect (see. Fig.



1and supplementary material C). All profiles were excavated in alluvial deposits in order to minimise the effects of different depositional processes on the profiles (cf. Fig. 2, Fig. S5). The profile depths varied between 35 cm (**P5**) and 80 cm (**P4**) (cf. Fig. 3, Table S1). Luminescence samples were collected under an opaque black plastic tarp under red light conditions. The

sample material was carved out of the profiles into opaque black plastic bags. Each profile unit was further sampled under daylight conditions for dose rate determination. The retrieved samples, including their depths, are listed in Table S1. Additionally, we used dust samples that were collected in an earlier study, from four different dust traps in the northern Atacama Desert installed at a height of 2 m (Wennrich et al., 2024).

**Figure 2: Schematic structure of an alluvial fan system in the Atacama Desert (modified from Walk, 2020, after Blair and McPherson, 2009). The sampling location of sampled alluvial fans and ephemeral channels are indicated by coloured boxes and profile names (P1–P5). The abandoned left segment of the terminal alluvial fan is dominated by sheet floods, whereas the right abandoned segment and the active depositional lobe are dominated by debris flows.**





**Figure 3: Photographs of the five investigated sampled profiles. (a) Profile P1 (103 m a.s.l.), (b) profile P2 (577 m a.s.l.), (c) profile P3 (1310 m a.s.l.), (d) profile P4 (1480 m a.s.l.), and (e) profile P5 (1930 m a.s.l.). White circles indicate the depth of samples used for all analysis and coloured circle the depth of samples only analysed for their geochemistry and granulometry. The individual lithological units within each profile are marked by dashed lines and labelled using Roman numerals and are described in supplementary material C. Profile units apply only to the respective profile and are not intended for correlations between profiles.**




## 3.2 Geochemistry and granulometry

Geochemical and granulometric analyses were conducted to characterise the sampled profiles and to identify potential indicators of depositional processes, sediment transport mechanisms, and pedogenic overprints. The material used for the analyses was generally taken from the luminescence samples. Where the available material was insufficient, additional samples for geochemical and sedimentological analyses were collected from the same sediment profiles but from different depths. Consequently, not all depths of the geochemical and sedimentological samples correspond exactly to those of the luminescence samples (cf. Table S1). Elemental concentrations were derived using the energy-dispersive X-ray spectrometer SPECTRO XEPOS (SPECTRO Analytical Instruments Ltd.) and for granulometric analyses a laser diffraction particle size analyser (Beckman Coulter LS13320) was used. For further details on sample preparation, measurement settings, and data analysis see supplementary material A.

## 3.3 Feldspar luminescence dating

The luminescence samples were prepared in the Cologne Luminescence Laboratory (University of Cologne; CLL) under subdued red-light conditions. As the quartz from the Atacama is not suitable for luminescence dating mainly due to its insufficient luminescence sensitivity (e.g., Bartz et al., 2020a; Del Río et al., 2019; Zinelabedin et al., 2022) the attention was shifted to feldspar luminescence dating. Automated Risø TL/OSL readers DA-20 (Bøtter-Jensen et al., 2010) were used for single-grain measurements of the 200 – 250 μm fraction of feldspar separates. A single-aliquot regenerative-dose (SAR) post-infrared infrared stimulated luminescence (pIRIR) protocol (Thomsen et al., 2008), adapted for single grains (Reimann et al., 2012), with a preheat of 250 °C for 60 s, an IR stimulation temperature of 50 °C for 2 s, and a pIRIR stimulation temperature of 225 °C for 3 s ($pIRIR_{225}$; Buylaert et al., 2009) was used. All equivalent dose ($D_e$) estimates were calculated using the numOSL R package version 2.8 (Peng et al., 2018). A 2 % measurement error was used for the regenerative dose signals ($L_x$) and the corresponding test dose signals ($T_x$). The $pIRIR_{225}$ signal exhibits generally low fading rates, ranging from $-0.7 \pm 0.1$ % to $1.9 \pm 1.4$ % per decade, with a mean fading rate of $1.1 \pm 0.1$ % per decade ($\pm$ standard error, n = 21). Following Roberts (2012), no fading correction was applied to the $pIRIR_{225}$ ages.

Uranium (U), thorium (Th), and potassium (K) concentrations were determined using an Ortec Profile MSeries GEM Coaxial P-type high-precision Germanium Gamma-Ray detector. The internal K-concentration was determined using a Risø GM beta multicounter system (Bøtter-Jensen and Mejdahl, 1985) and a Bruker M Tornado μ-XRF following the approach proposed by Maßon et al. (in press). Dose rates were calculated using the Dose Rate and Age Calculator (DRAC, Durcan et al., 2015). For further details on sample preparation, measurement protocol and its determination, OSL reader specification, luminescence data analysis, and dose rate calculations see supplementary material B.

To obtain information about the depositional age of the alluvial deposits and post-depositional activity in the profiles, we applied different age models (Galbraith et al., 1999), which are implemented in the R Luminescence package version 0.9.25 (Burow, 2024a, 2024b; Kreutzer et al., 2012). Given the high dispersion in our $D_e$ distributions caused by either heterogeneous



bleaching and/or post depositional mixing, we applied the 4-parameter Minimum Age Model (MAM4) to estimate the minimum dose (Galbraith et al., 1999; Galbraith and Roberts, 2012). When the MAM4 failed to converge and provide a reliable estimate, we used the more robust 3-parameter MAM (MAM3) instead. In cases where both models were unable to provide a result, or when the models estimated that the fraction of grains contributing to the minimum dose component is very small (<10 %), we opted for a Central Age Model (CAM). Due to the occurrence of negative doses in the samples, we applied all age models on unlogged data (Arnold et al., 2009). The implementation of the unlogged MAM in the R Luminescence package requires a single value of natural overdispersion, called sigma-b ($\sigma_b$), in measured units (Gray) for all grains in a sample. As this will lead to a disproportionally large error for grains with near-zero doses and a negligible effect for grains with very high doses, we opted to add $\sigma_b$ on a grain level instead of on a sample level. The grain-wise error was calculated by adding a $\sigma_b$ of 30 % of the equivalent dose to the measurement error in quadrature. This approach is consistent with the logged version of the MAM, where the relative $\sigma_b$ is added to each individual relative measurement error.

To determine the final depositional age or the time of the last post-depositional mixing activity, we assessed whether i) the measured grains can be effectively bleached, meaning they do not retain significant residual doses after sunlight exposure, and ii) the bleaching depth does not extend far into the profile. Since the surface sample of each profile was taken from the uppermost two centimetres, it represents a mixture of grains exposed to sunlight and grains with up to two centimetres of overburden. These surface samples do not provide information about recent activity but instead serve as indicators of the natural bleaching characteristics of the profile. In addition, laboratory bleaching tests were conducted on a multi-grain and single-grain level using a SOL2 solar simulator for 24 h.

We further calculated two additional proxies to detect potential post-depositional mixing processes. These are i) the fraction of grains with a saturated luminescence signal (Reimann et al., 2017), and ii) the fraction of grains with $D_e$ values near zero (Bateman et al., 2007). Saturated grains, originating from the bedrock or deposits older than the luminescence dating range, indicating upward transport when found in younger sediments in soil profiles. However, in late Pleistocene to Holocene alluvial deposits, their interpretive value appears to be more limited, as deeper depositional units may not be in saturation. Conversely, grains with $D_e$ values near zero in older deposits indicate downward transport (Bateman et al., 2007; Gliganic et al., 2016). We modified the "zero-dose" grains proxy by Bateman et al. (2007) and defined it as grains that agree within one standard deviation with the MAM4 minimum dose of the surface samples (<2.24 Gy). Because our surface samples retain a residual dose from incomplete bleaching, a common property of feldspar grains, we use this value instead of a true zero dose. The use of this proxy is limited in recently deposited units, where the dose indicates the timing of deposition rather than downward transport, and if doses >2.24 Gy have relatively large errors.



# 4 Results and interpretation

## 4.1 Geochemistry and granulometry

Most samples are sand-dominated. Only the samples **P4-1** and **P4-2** are dominated by very coarse silt (Fig. 4a). The dust

samples (cf. Wennrich et al., 2024) are either mud or sandy mud, with comparatively low sand fractions of 3.2 % to 12.7 %

(Fig. 4a). All samples are either poorly or very poorly sorted according to Folk and Ward (Blott and Pye, 2001; Folk and Ward,

1957). Exemplary grain size distributions from each profile and a mean distribution of the dust samples are shown in Fig. 4b.

For detailed results on the grain size distributions see Fig. S2.

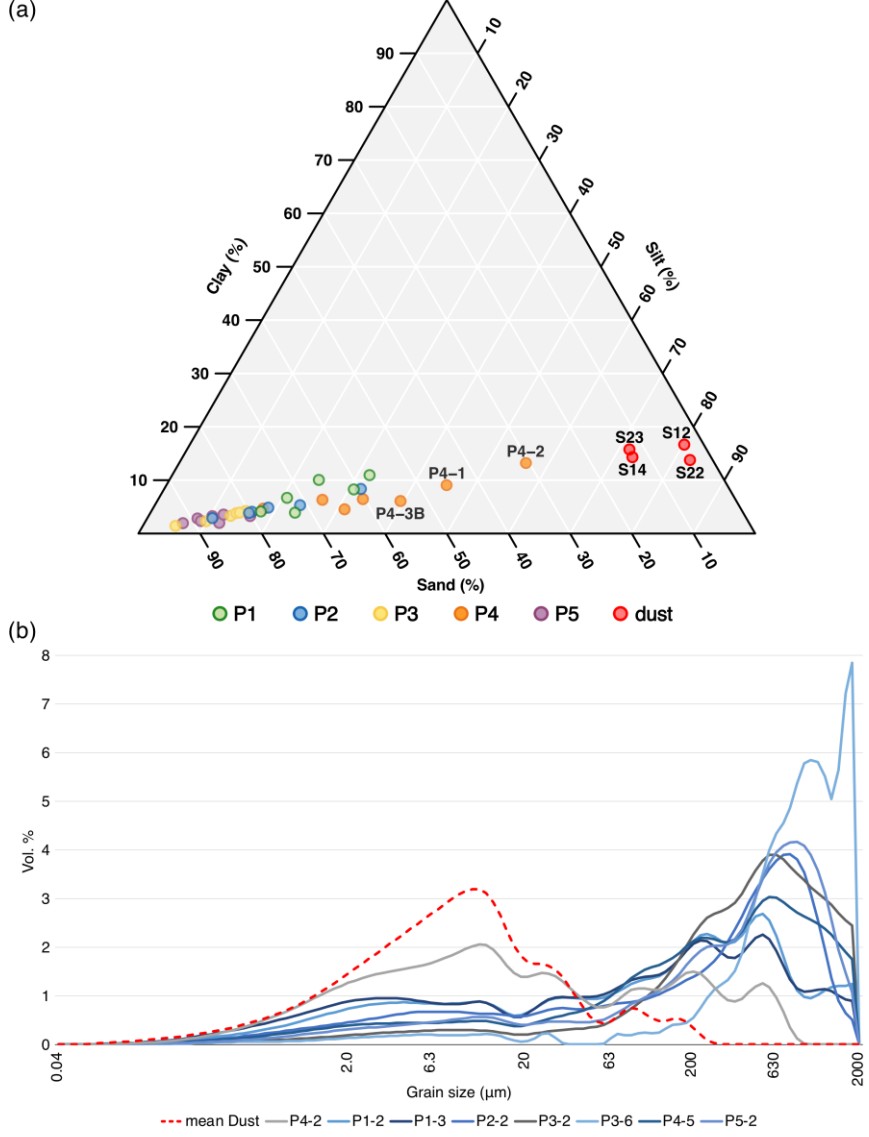

**Figure 4: (a) Soil texture triangle of all sediment samples and the four dust samples, and (b) grain size distributions of exemplary**

**samples including the mean values for the four dust samples (dotted line).**





Since common weathering indices are not suited for application in deserts (Chen et al., 2021), a ratio of mobile (Na, Mg, K, Ca) to immobile (Al, Ti) elements was calculated based on elemental concentrations obtained by XRF analysis ($\sum E_m/\sum E_{im}$, Fig. 5a). In general, smaller ratios indicate leaching of mobile elements. Due to the geochemical differences in the parent

material, a comparison between different profiles is not possible. The biggest intra-profile differences occur in **P1** and **P4**.

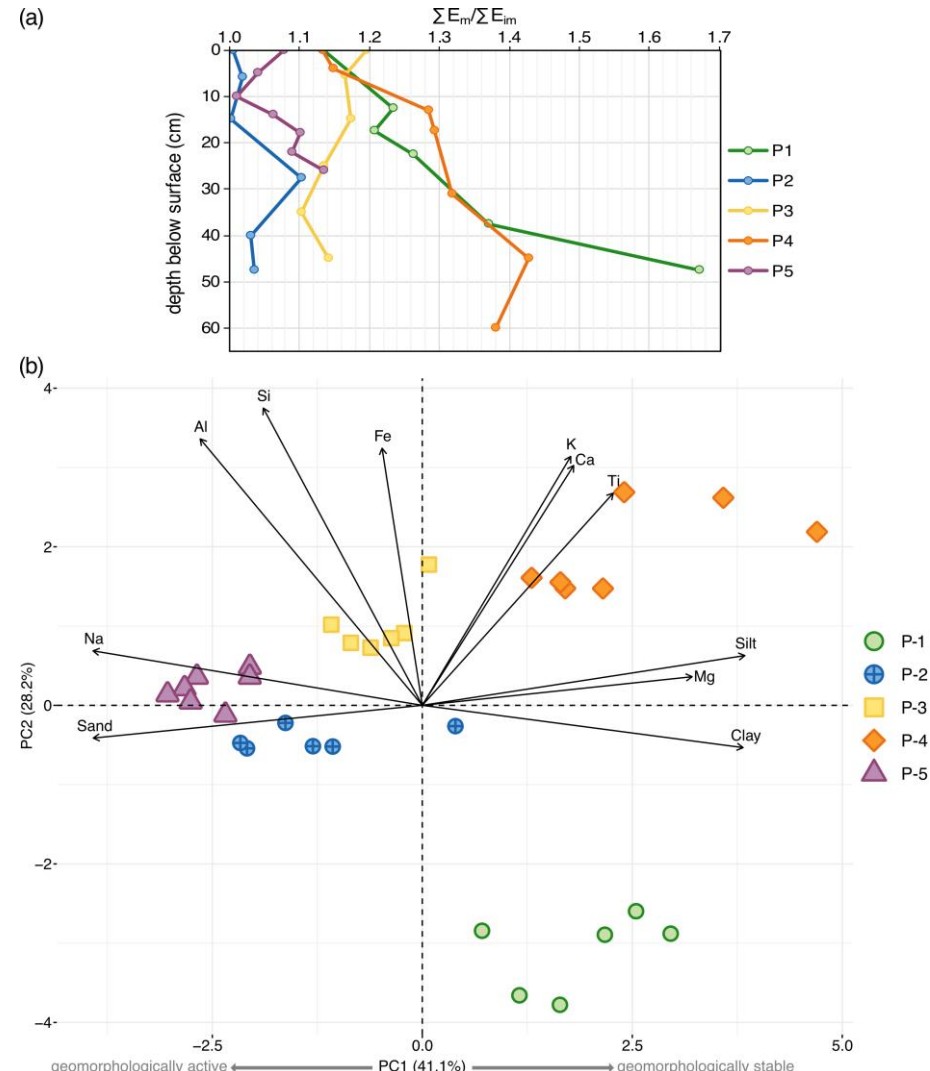

**Figure 5: (a) Mobile to immobile ratio ($\sum E_m/\sum E_{im}$) against depth. Smaller ratios indicate the leaching of mobile elements and therefore weathering. (b) PCA of all sediment samples based on grain size parameters and elemental concentrations.**

To allow for a comparison of the geochemical composition and granulometric results among the profiles and to assess the

degree of weathering, a principal component analysis (PCA) was conducted. The input data included grain size parameters (sand, silt, and clay fractions), relatively immobile and weathering-resistant elements (Al, Ti), relatively mobile and



weathering-susceptible elements (K, Na, Ca, Mg, Si), as well as the Fe content (Fig. 5b). The first two principal components (PC1 and PC2) explain 41.1 % and 28.2 % of the total variance, respectively, thereby accounting for ~70 % of the variability of the dataset. While PC1 is primarily influenced by the granulometric properties, PC2 is influenced by the geochemical

components. The PCA reveals a distinct separation of samples by profile rather than by degree of weathering. Samples from **P1** and **P4** plot in the positive PC1 half, associated with higher Mg-concentrations and fine grain (silt and clay) contents. Along PC2, the samples of **P1** (negative half) and **P4** (positive half) are primarily separated due to their differing Al-, Si-, Fe-, K-, Ca-, and Ti- concentrations. Samples of **P2**, **P3**, and **P5** plot mostly in the negative half of PC1, with the exceptions of sample **P2-3B** (no equivalent luminescence sample available) and **P3-4**. Their separation along PC2 is primarily caused by their

differing concentrations of Al, Si, Fe, K, Ca, and Ti.

### 4.2 Feldspar luminescence ages

All surface samples (**P1-1**, **P2-1**, **P3-1**, **P4-1**, and **P5-1**) show a mixture of recently bleached grains <5 Gy and grains with $D_e$ >50 Gy (Fig. S6-10). This indicates that under natural conditions i) grains at the immediate surface are sufficiently bleached, and ii) bleaching ceases already at very shallow depths, i.e., within the uppermost 2 cm. Therefore, in the following (sections

4.3 - 5.2 and the figures therein), only the ages of the underlying sedimentary units within each profile are discussed, yet all ages and dose rate-relevant elemental concentrations can be found in Tables 2 and S5. On average, residuals in the laboratory bleaching tests were found to be <2 Gy for multi-grain analysis and <1 Gy for single-grain tests, which further demonstrates the potential of the samples to be effectively bleached (Fig. S3e, Fig. S4a-g).

The single-grain pIRIR$_{225}$ $D_e$ distributions of most samples are broad, spanning from 0 Gy up to saturation in some cases (Fig

6). While samples close to the surface show a unimodal right skewed distribution, deeper samples in the profiles **P2**, **P3**, and **P5** display more complex distributions. The total dose rates range from 2.41 Gy ka$^{-1}$ (**P5-2**) to 5.00 Gy ka$^{-1}$ (**P4-4**). **P2** and **P5** have the lowest and **P4** the highest dose rates. Sub-surface zero-dose grains range from 0 % to 66.2 %, usually gradually decreasing downward (Table 2). Sub-surface saturated grains range from 0 % to 47.2 %, usually gradually increasing downward (Table 2).



**Table 2: Luminescence results, including the dose model used for calculation of the $D_e$, the $D_e$, the total dose rate ($\dot{D}$), the age, the fraction of zero-dose grains and the fraction of saturated grains per sample.**

| Sample | Unit (profile-wise) | Model used | $D_e$ (Gy) | $\dot{D}$ (Gy ka$^{-1}$) | Age (ka) | Zero-dose grains (%) | Saturated grains (%) |
|---|---|---|---|---|---|---|---|
| **Profile P1** | | | | | | | |
| **P1-1** | I | MAM-4 | 1.60 ± 0.17 | 3.85 ± 0.50 | 0.42 ± 0.07 | 79.5 | 0 |
| **P1-2** | II | MAM-4 | 10.9 ± 1.45 | 3.91 ± 0.62 | 2.78 ± 0.57 | 2.07 | 3.45 |
| **P1-3** | III | CAM | 106 ± 5.02 | 3.77 ± 0.59 | 28.07 ± 4.55 | 0 | 3.37 |
| **P1-4** | IV | CAM | 107 ± 2.77 | 3.71 ± 0.69 | 28.71 ± 5.37 | 0.64 | 2.88 |
| **Profile P2** | | | | | | | |
| **P2-1** | I | MAM-4 | 1.52 ± 0.55 | 2.35 ± 0.25 | 0.65 ± 0.24 | 74.2 | 1.08 |
| **P2-2** | II | MAM-4 | 1.05 ± 0.57 | 2.50 ± 0.28 | 0.42 ± 0.23 | 66.2 | 1.54 |
| **P2-3** | II | MAM-4 | 2.73 ± 0.46 | 2.54 ± 0.29 | 1.08 ± 0.22 | 56.2 | 0 |
| **P2-4** | IV | MAM-4 | 1.83 ± 1.02 | 2.52 ± 0.35 | 0.73 ± 0.42 | 33.3 | 0 |
| **Profile P3** | | | | | | | |
| **P3-1** | I | CAM | 4.93 ± 0.63 | 3.16 ± 0.35 | 1.56 ± 0.26 | 43.2 | 2.11 |
| **P3-2** | II | MAM-4 | 2.94 ± 0.74 | 3.36 ± 0.51 | 0.87 ± 0.26 | 19.8 | 5.81 |
| **P3-3** | II | MAM-4 | 3.96 ± 1.11 | 3.43 ± 0.54 | 1.16 ± 0.37 | 10.2 | 7.63 |
| **P3-4** | III | CAM | 6.31 ± 5.68 | 3.43 ± 0.54 | 1.84 ± 1.68 | 9.78 | 9.78 |
| **P3-5** | IV | CAM | 184 ± 12.2 | 3.48 ± 0.59 | 53.0 ± 9.59 | 1.12 | 30.3 |
| **P3-6** | IV | CAM | 208 ± 12.8 | 3.60 ± 0.59 | 57.9 ± 10.07 | 0 | 36.7 |
| **Profile P4** | | | | | | | |
| **P4-1** | I | MAM-4 | 4.72 ± 0.26 | 4.28 ± 0.49 | 1.10 ± 0.14 | 30.3 | 0.90 |
| **P4-2** | II | CAM | 29.5 ± 2.69 | 4.51 ± 0.56 | 6.55 ± 1.01 | 1.33 | 10.0 |
| **P4-3** | III | CAM | 205 ± 13.0 | 4.92 ± 0.72 | 41.7 ± 6.65 | 0.72 | 46.4 |
| **P4-4** | V | CAM | 224 ± 10.6 | 5.00 ± 0.70 | 45.0 ± 6.65 | 1.52 | 41.4 |
| **P4-5** | VII | CAM | 227 ± 11.9 | 4.53 ± 0.89 | 50.1 ± 10.17 | 0.62 | 47.2 |
| **Profile P5** | | | | | | | |
| **P5-1** | I | MAM-3 | 2.92 ± 0.34 | 2.26 ± 0.25 | 1.29 ± 0.21 | 58.0 | 2.76 |
| **P5-2** | II | MAM-4 | 3.42 ± 0.44 | 2.41 ± 0.28 | 1.42 ± 0.25 | 50.5 | 1.77 |
| **P5-3** | III | MAM-3 | 4.39 ± 0.59 | 2.48 ± 0.29 | 1.77 ± 0.31 | 36.6 | 3.23 |
| **P5-4** | IV | MAM-4 | 9.65 ± 1.07 | 2.44 ± 0.35 | 3.95 ± 0.71 | 15.6 | 8.89 |
| **P5-5** | IV | MAM-3 | 10.8 ± 1.49 | 2.46 ± 0.35 | 4.41 ± 0.87 | 7.58 | 4.55 |
| **P5-6** | V | MAM-4 | 45.0 ± 3.42 | 2.52 ± 0.39 | 17.9 ± 3.06 | 0 | 7.61 |










**Figure 6: D$_e$ distributions and corresponding MAM or CAM D$_e$ (left) and ages (right) of all five sampled profiles, excluding the surface samples. Ages from earlier studies referenced in the main text, are shown as dotted red lines. Their conversion into D$_e$ was based on an average dose rate of the corresponding profile. The x-axis for the D$_e$ distributions is truncated at 500 Gy to facilitate visualization.**

## 4.3 Interpretation of profile genesis

**Profile P1 – 103 m a.s.l.**

**P1** is located on the surface of the oldest depositional generation (Q1, Walk et al., 2023) of a large coastal alluvial fan just south of Paposo (supplementary material C; Fig. S6). The deposition of this alluvial fan generation was previously dated to $111 \pm 20$ ka (i.e., MIS 5) based on $^{10}$Be exposure ages of boulders at the fan surface (Walk et al., 2023). Based on various soil proxies, these authors also suggested that weathering processes on this surface have resulted in the evolution of a well-developed soil horizon. A clear downward trend is also evident in our geochemical data for **P1**, suggesting that mobile elements were depleted from the top layers, as generally expected for top soil layers (Fig. 5a). Further, the silt and clay fractions of units II, III and the upper part of IV are slightly increased, which potentially indicates initial weathering and soil forming processes as well (Fig. 4a, Fig. S6). Unit I is the only one extensively penetrated by roots, with diameters reaching up to 5 mm while in unit II, only a few fine roots are present, and their abundance decreases with depth. Starting from unit III, no roots are observed. Yet, none of the grains in sample **P1-2** and merely 2.3 % of the grains in sample **P1-3** respectively 4.6 % in sample **P1-4** have a D$_e$ in agreement with the depositional age ($111 \pm 20$ ka = ~$420 \pm 75$ Gy) according to Walk et al. (2023). Recent to subrecent material exchange between the surface and deeper profile units IV and V is unlikely, as zero-dose grains are absent or rare (<1 %) in samples **P1-3** and **P1-4** (Table 2). Moreover, the D$_e$ peak in **P1-2** (CAM D$_e$ 50.1 Gy) is largely absent in the distributions of **P1-3** and **P1-4**, and only minor portions of their peaks appear in the right shoulder of **P1-2**. We therefore interpret that **P1-3** and **P1-4** belong to a depositional phase around ~28 ka, rather than being the result of post-depositional vertical mixing. However, in the uppermost ~10 cm, post-depositional mixing may be inferred from the D$_e$ distribution of sample **P1-2**. Although **P1-2** contains only 2 % zero-dose grains, the shoulder on the left flank of the distribution might represent material mixing in the form of vertical grain transport from the surface (**P1-1**) to the depth of **P1-2**. This portion of the D$_e$ distribution of **P1-2** is reflected in the MAM age of $2.78 \pm 0.56$ ka indicating that mixing processes were active at least until that time. The CAM age, in contrast, indicates a depositional phase around 13 ka for sample **P1-2**. Bioturbation caused by the observed roots and the initial soil processes indicated by our granulometric and geochemical analysis support this interpretation. The discrepancy between the luminescence inferred depositional age presented here, and the cosmogenic nuclide-based ages reported by Walk et al. (2023) may be explained by: i) a younger phase of fluvial reworking or resurfacing (cf. Haug et al., 2010) on top of the older alluvial fan generation; ii) cut-and-fill processes and lateral sedimentation associated with the formation of the fourth and youngest alluvial fan generation inside the main channel, which may have reworked or reactivated surface sediments of the oldest fan surface, incorporating younger material; or iii) inherited age in the cosmogenic nuclide ages reported by Walk et al. (2023).



**Profile P2 – 577 m a.s.l.**

The profile is located within the marine boundary layer and thus in a zone of high fog occurrence and abundant loma vegetation
(Cereceda et al., 2008; del Río et al., 2018; supplementary material C; Fig. S7). Unit I of profile **P2** is penetrated by both fine
and thick (~2 cm in diameter) roots, extending into unit II. No roots are observed in unit III, yet fine roots occur again in
underlying unit IV. The grain size analysis shows an overall downward increase of grain size and only little variation is visible
in the geochemical parameters (Fig. 4a, Fig. 5a, Fig S6). The $D_e$ distributions of samples **P2-2** and **P2-3** show a peak in a low
dose range and a short tail towards the higher doses (>25 Gy) (Fig. 6). Saturated grains are virtually absent (Table 2), which
might indicate the recent deposition of heterogeneously bleached grains and/or intense post-depositional mixing processes. A
combination of deposition and post-depositional mixing seems to agree with i) the presence of an active fault line (Paposo
fault) in the direct adjacency, and ii) the location within the marine boundary layer that is presumably linked to higher moisture
availability and rainfall frequency causing increased alluvial activity compared to higher elevations and enabling bioturbation
as a mixing process. The MAM4 ages of samples **P2-2** (0.42 ± 0.23 ka) and **P2-3** (1.08 ± 0.22 ka) and their position within the
same unit (unit II) suggest deposition of the upper profile within the last ~1200 years. Additional post-depositional mixing is
indicated by i) a decreasing fraction of zero-dose grains from 74 % (**P2-1**) to 56 % (**P2-3**) (Table 2), and ii) a widening of the
$D_e$ peak bases from **P2-1** to **P2-3** (Fig. 6, Fig. S7). The downward decline in zero-dose grains suggests a depth-dependent
reduction in post-depositional vertical material mixing processes, likely caused by root-related bioturbation. This aligns with
the lithology, as unit II appears heterogeneous and contains numerous roots. The broader $D_e$ peaks in **P2-2** and **P2-3** likely
reflect mixing with adjacent sediment layers, where more old grains are incorporated in proximity to older underlying layers.
The high root density supports bioturbation as the dominant process driving this vertical grain movement.

The lowermost sample **P2-4** (unit IV) contains a distinct older grain population (CAM 11.0 ± 2.7 ka) in addition to a $D_e$ peak
consistent with the overlying samples (MAM 0.73 ± 0.42 ka, Fig. 6) suggesting possible synchronous deposition. The older
grain population could reflect intensive post-depositional mixing or reworking of older sediments during the deposition of all
four units. However, since i) **P2-4** represents the uppermost 10 cm of unit IV, and ii) roots are present in unit IV but absent in
unit III, suggesting limited vegetation cover of unit IV and thus a paleosurface, we interpret unit IV as being deposited >11 ka.
The younger $D_e$ peak likely reflects the paleosurface age, and thus the burial of unit IV and deposition of unit III ~700 years
ago.

**Profile P3 – 1310 m a.s.l.**

Similar to **P2**, the grain size of the brown-yellowish matrix in **P3** shows an overall downward increase (supplementary material
C; Fig. S8). All units are dominated by the coarse to very coarse sand fractions, and the mobile to immobile element ratio
reveal minimal differences between the individual profile units (Fig. 4a, Fig 5a). Although located in elevations at or above
the upper limit of the marine boundary layer, the MAM and CAM ages from the uppermost units in **P3** (**P3-2**, **P3-3**, **P3-4**)
suggest that the last bleaching activity, likely related to deposition, at **P3** took place during the past 2000 years, similar to **P2**





(Fig. 6). In contrast to **P2**, however, the single-grain $D_e$ distributions of these uppermost samples of **P3**, including surface sample **P3-1**, are very broad (Fig 6, Fig. S8). Broad peaks together with the occurrence of saturated grains and zero-dose grains, suggest heterogeneous bleaching and the erosion and reworking of surface sediments as well as previously buried older material during deposition. Below, samples **P3-5** and **P3-6** display broad unimodal $D_e$ distributions likely indicating a rather long burial time. Together with the high proportion of saturated grains (>30 %) and the absence of surfaced grains ($\leq$1 %)

(Table 2), the CAM ages of samples **P3-5** and **P3-6** provide evidence of a much older depositional period or event at **P3** that took place around 55 ka (**P3-5**, 53.0 ± 9.6 ka; **P3-6**, 57.9 ± 10.1 ka).

**Profile P4 – 1480 m a.s.l.**

**P4** is located on the surface of an older alluvial fan generation earlier described by Moradi et al. (2020) and Sun et al. (2023), several 100 metres above the upper limit of the marine boundary layer and thus outside the zone of frequent fog occurrence

and loma vegetation (supplementary material C; Fig. S9). As in **P2** and **P3**, profile **P4** shows an increase in average grain size with depth, transitioning from a silt-dominated upper unit to sand-dominated layers below (Fig. 4a, Fig. S9). The mobile to immobile element ratio likewise increases, particularly from unit II (**P4-2**) to unit III (**P4-3**), whereas only small variations are visible in the deeper units III-VII (Fig. 5a). The $D_e$ distribution of **P4-2** is right skewed, peaking around 30 Gy (Fig. 6). For this sample, a MAM age could not be calculated and the CAM age (10.6 ± 1.5 ka) corresponds to the beginning of the shoulder

on the right flank rather than the peak or leading edge. The peak mode is ~29.5 Gy (6.6 ka), and the leading edge is estimated at ~13 Gy (3 ka) (Fig. 6). Below, samples **P4-3** to **P4-5** are characterised by i) the virtual absence of zero-dose grains (<2 %), ii) >40 % of saturated grains, and 3) broad unimodal $D_e$ distributions, altogether likely indicating a long-term of burial (Fig. 6). Saturated grains presumably reflect incomplete bleaching due to the steep and confined catchment area of **P4**.

The absence of zero-dose grains and the broad unimodal distribution of the lower samples P4-3 to P4-5 suggest that the CAM

ages of these samples represent a depositional period between 50.1 ±10.2 ka (CAM **P4-5**) and 41.7 ± 6.7 ka (CAM **P4-3**). Those CAM ages agree within 2σ with the proposed depositional age of 56.4 ± 2.8 ka for this fan generation calculated by Sun et al. (2023) (Fig. 6). In addition, these authors date the deposition of the adjacent younger fan generation to ~13.6 ± 1.8 ka, which agrees with the CAM age of 10.6 ± 1.5 ka calculated for sample **P4-2** that was taken from the uppermost sub-surface unit in P4. However, surface characteristics at **P4** (i.e., varnished and rubified clasts, planar topography, partly indurated

bedding) rather suggest a longer period of depositional inactivity when compared to the younger fan section that is indicated by grey clasts, a distinct bar-and-swale topography and loose bedding (Sun et al., 2023). To explain the young CAM age and long tail of the distribution of **P4-2**, it may be assumed that unconcentrated sheet wash events, likely contemporaneous to the alluvial activity in the younger fan generation, have led to the successive accumulation of fine-grained unit II on top of the older fan section. Additionally, processes related to the evolution of desert pavements , i.e. the trapping of fine sediment below

the gravel-covered surface (aeolian & sheet wash) may explain the young peak including its left shoulder (Dietze et al., 2016; Ugalde et al., 2020; Williams and Zimbelman, 1994). **P4-2**'s grain size distribution also more closely resembles that of dust samples from northern Atacama than any other sample (Fig. 4b), supporting this hypothesis. The $D_e$ distribution may thus



reflect intensified dust trapping and desert pavement development between ~7 and 3 ka that occurred on top of an older (40-50 ka) alluvial fan surface.

The absence of both peak and assumed leading edge of the $D_e$ distribution of **P4-2** (~30 Gy, ~13 Gy) in underlying $D_e$ distributions suggest little to no vertical mixing between units II and III. In addition, the very low zero dose fraction in **P4-2** (<2 %) further implies negligible downward transport from **P4-1** at present (Table 2). Accordingly, present day post-depositional mixing at **P4** appears limited, which agrees with the fact that vegetation-related bioturbation and maybe even desert pavement evolution is impeded under present environmental conditions.

**Profile P5 – 1930 m a.s.l.**

Overall, no consistent trend in grain size with depth is observed and all units are dominated by coarse to very coarse sand in **P5** (Fig. 4a, supplementary material C; Fig. S10). The ratio of mobile to immobile element does not vary significantly across the profile (Fig. 5a). All single-grain $D_e$ distributions show large scatter (Fig. 6). Samples **P5-2** to **P5-5** show unimodal, right-skewed $D_e$ distributions with progressively broader peaks towards the base of the profile, whereas sample **P5-6** lacks a clearly

defined peak. While the fraction of saturated grains is small throughout the entire profile (2-9 %) and not increasing consistently with depth, the fraction of zero-dose grains constantly decreases with depth from 50 % in **P5-2** to 0 % in **P5-6** (Table 2). However, due to the young MAM ages of the samples the zero-dose grains more likely reflect recent deposition rather than post-depositional mixing. The MAM ages of the upper two sample **P5-2** (1.42 ± 0.25 ka) and **P5-3** (1.77 ± 0.31 ka) from units II and III agree within 1σ, indicating a deposition during the last 2 ka like the upper units of **P2** and **P3** (Fig. 6).

Unit IV likely belongs to a period of deposition between ~5 and 3 ka (**P5-4,** 3.95 ± 0.71 ka; **P5-5,** 4.41 ± 0.87 ka, Fig. 6). In addition, sample **P5-6** from unit V displays a markedly different $D_e$ distribution compared to the overlying samples (Fig. 6). Its MAM age (17.9 ± 3.1 ka) also deviates significantly from the overlying samples. Based on our ambiguous data, we cannot identify the processes responsible for the $D_e$ distributions in sample **P5-6**, and therefore cannot interpret the age of unit V.

**5 Discussion**

**5.1 Profile chronology and formation**

The luminescence data obtained from all five profiles reveal characteristic patterns of both depositional and post-depositional processes. In all profiles, the single-grain $D_e$ distributions range from narrow unimodal to very broad and occasionally multimodal shapes, suggesting varying degrees of bleaching prior to deposition and post-depositional disturbances (Fig. 6). Alluvial deposits are generally prone to heterogeneous bleaching of the luminescence signal, since they are usually the result

of fast but intense discharge events with high sediment-to-water ratios that prohibit bleaching of all sediment (Duller, 2008; Ventra and Clarke, 2018). In this context, it is therefore important to perform single-grain analyses, as the averaging effect of single-grain signals in multi-grain measurements can obscure important age components (Duller, 2008). We observe $D_e$ distributions with a distinct peak around a low dose ~2.24 Gy and a long tail reaching into a higher dose region >50 Gy in all



our surface samples as well as in samples **P2-2** and **P2-3** (cf. Fig. 6, Fig. S3), which is a typical characteristic of heterogeneous

bleaching in fluvial transport (Guibert et al., 2017) (Fig. 7b).

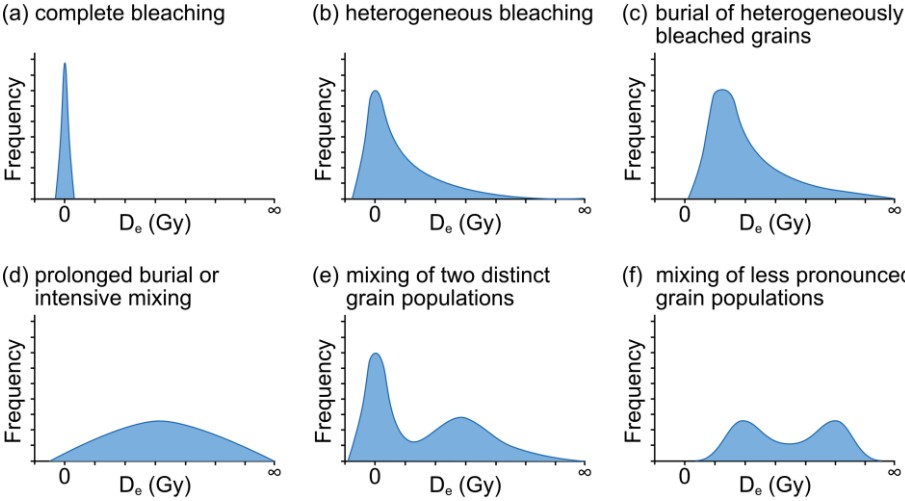

**Figure 7: Theoretical $D_e$ distributions adapted from Bateman et al. (2003): (a) complete bleaching with a narrow normal distributed $D_e$ distribution around a dose of 0 Gy, (b) heterogeneous bleached grains with a distinct peak in a low dose region and a tail towards higher doses, (c) heterogeneous bleached grains after burial, shifting the low dose peak to the right and broadening it, (d) prolonged**
**burial or intensive mixing of heterogeneous grain populations represented by a unimodal normal distributed $D_e$ distribution, (e) mixing of two distinct grain populations resulting in a bi-modal distribution with the older grain population being broader than the recently buried one, (f) mixing of two less pronounced grain populations.**

The length of the tail represents the inherited dose that was accumulated in the individual grains prior to alluvial transport and the associated bleaching period. In samples **P3-2** to **P3-4**, **P4-2**, and **P5-2** to **P5-5**, the $D_e$ distributions likewise exhibit a long

tail to the right yet their peaks are broader and in slightly higher dose regions (Fig. 6 and 7c). This indicates a burial phase of heterogeneous bleached samples, where the entire distribution will shift towards higher doses and will disperse due to differences in micro-dosimetry and mixing (Bateman et al., 2003; Cunningham et al., 2015, Fig. 7c). Broad unimodal distributions are observed in samples **P1-2** to **P1-4**, **P3-5**, and **P4-3** to **P4-5** (Fig. 6 and 7d). Those broad unimodal distributions suggest either prolonged phases of burial, where effects like micro-dosimetry and/or mixing lead to a substantial broadening

of the peak, or intense mixing after burial (Fig. 7c). In turn, if a heterogeneous bleached sediment layer remains at the surface and therefore experiences bleaching over a longer time period, the peak will become more pronounced and the tail will vanish over time (Duller, 2008; Fig. 7a). However, in this study we took surface samples ranging up to 2 cm in depth. Therefore, we incorporated grains into our surface samples that have not been exposed to light at the present surface. Since the $D_e$ distributions of all surface samples have grains reaching into high doses (~100 Gy for **P1-1** up to ~600 Gy for **P5-1**, Fig. S6-S10 similar to

Fig. 7b) we can infer that the bleaching depth within Atacama Desert soils and surfaces is less than 2 cm. Nevertheless, almost 85 % of the grains in the surface samples can be considered as well bleached (p0 of the MAM4 of all surface samples = 0.85). Finally, bimodal distributions occur in samples **P2-4, P3-6**, and **P5-6** (Fig. 6). Bi- or multimodal $D_e$ distributions may indicate post-depositional mixing processes, such as bioturbation caused by roots or burrowing organisms, resulting in the appearance





of one or more additional peaks alongside the original heterogeneously bleached component (Bateman et al., 2003; Fig. 7e and
f). Alternatively, such distributions may also reflect the input of sediment from multiple sources with distinct inherent
luminescence signals that were not reset during deposition.

The different shapes of $D_e$ distributions offer insights into depositional and post-depositional processes on a sample level. Yet,
it is not possible to definitively determine whether each sample was shaped by either depositional or post-depositional
processes. For their interpretation, $D_e$ distributions should always be accompanied by stratigraphy, field observations, sediment
properties and additional proxies such as saturated and zero-dose grains. Unfortunately, the PCA based on geochemical and
granulometric parameters mainly provided site-specific characteristics (Fig. 5b), indicating that sediment source and local
environmental conditions have a stronger effect on the sediment properties than weathering, post-depositional mixing or
depositional processes. Nevertheless, based on the combination of $D_e$ distributions, field observations, elemental ratios, grain
size properties and the PCA, we can distinguish two main types of profiles: i) profiles that are associated with
**geomorphologically active** surfaces (**P2**, **P3**, **P5**, left part of the PCA biplot, Fig. 5b), and ii) profiles that are associated with
**geomorphologically stable** surfaces (**P1** and **P4**, right part of the PCA biplot, Fig. 5b). The sediments and chronology of the
profiles associated to **geomorphologically active** surfaces are dominated by recent to sub-recent depositional processes,
mainly in the form of subrecent/late Holocene alluvial activity, while the **geomorphologically stable** surfaces are affected by
post-depositional *in-situ* weathering and mixing processes in the absence of active geomorphic processes, as they lie outside
of the subrecent or modern alluvial activity.

A key feature of geomorphologically active profiles is the relatively high proportion (10-66 %) of zero-dose grains (i.e., $D_e$
<2.24 Gy) up to depth of 30 cm or more (Table 2, Fig. S7, S8, S10). This suggests that, despite the dominance of geomorphic
processes, there is an incorporation of recently bleached grains in the subsurface layers, either due to post-depositional mixing,
or due to erosion of modern surfaces and/or young depositional units within the catchment during deposition. The root
penetration in **P2** indicates a post-depositional mixing component, while the supposed paleosurface (**P2-4**) indicates recent
deposition of the upper three profile units. Given the absence of vegetation at the sites of **P3** and **P5**, bioturbation as a post-
depositional process can be excluded for these profiles. The downward decrease in zero-dose grains in these profiles is thus
more likely the result of mixing during recent deposition. During alluvial deposition, turbulent sediment-laden flows can
rework the pre-existing surface, driving recently bleached grains downward and causing their incorporation into underlying
sediments.

In contrast to geomorphologically active profiles, the stable profiles were sampled from older alluvial fan surfaces. In both
fans, the older surfaces are lying on higher terrace-type levels than the recent depositional activity, creating a topographic
barrier. **P1** is located in the coastal plain, which has a considerably higher relative air humidity (Table 1) and – though still
hyperarid – generally receives higher precipitation amounts compared to the Coastal Cordillera above the marine boundary
layer. These local climatic conditions promote weathering and vegetation growth, which in turn enhances bioturbation. In
contrast, **P4** is positioned in the arid Coastal Cordillera above the fog-influenced zone without the presence of vegetation.



Mixing in **P4** likely results from desert pavement dynamics under persistently hyperarid conditions (Dietze et al., 2016; Ugalde et al., 2020; Williams and Zimbelman, 1994). Although both profiles are situated in distinctly different climatic settings and are affected by different mixing processes, they exhibit evidence of post-depositional mixing restricted to the uppermost ~10

cm of the soil. This shallow mixing depth in both profiles is consistent with previous findings of limited downward translocation of salts and water-soluble colloids in Atacama soils (Ewing et al., 2006; Sun et al., 2023).

### 5.2 Paleoclimatic significance

Across the ~70 ka period covered by our data we identify two main phases of alluvial activity (Fig. 6). The first occurs around 50 ka during Marine Isotope Stage 3 (MIS 3) and is consistently recorded in profiles above the marine boundary layer (**P3**-

**P5**). The second phase of alluvial activity covers the last ~5 ka and is evident in the profiles from geomorphologically active settings (**P2**, **P3**, **P5**). The data also reveals three distinct intervals with no alluvial activity (Fig. 6). In the Coastal Cordillera, the first interval starts around 35 ka and ends at ~14 ka. In contrast, the profile in the coastal plain (**P1**) shows a phase of activity within this interval (~34–23 ka), effectively subdividing the period of alluvial inactivity into two shorter phases: a minor one from ~35–34 ka and a more extended one from ~23–14 ka (Fig. 6 right). The third period of alluvial inactivity is

evident across the entire transect from ~8.4 to ~5.3 ka (Fig. 6 right).

The first alluvial active phase at ~50 ka coincides with a marked reduction in regional aridity indicated by the aridity index of Stuut and Lamy (2004; Fig. 8a, g). It further aligns with previously reported activity patterns from alluvial deposits in the coastal plain located further north (Bartz et al., 2020a, 2020b; Vargas et al., 2006; Fig. 8f), suggesting a broader climatic control on sediment mobilisation during MIS 3. According to Stuut and Lamy (2004), this more humid phase reflects the effect

of precessional forcing of the Milankovitch cycle, which influenced the tropics and thereby the latitudinal position of Southern Westerlies.





**Figure 8: Regional proxies for humid phases: (a) Aridity Index ~27 °S (Stuut and Lamy, 2004), (b) humid phases based on colluvial sediments ~21 °S (Medialdea et al., 2020), colluvial aeolianite cementation ~28 °S (Nash et al., 2018), fluvial terraces ~21 °S (Gayo et al., 2023; Nester et al., 2007), (c) humid phases based on groundwater recharge ~22-25 °S (Rech et al., 2002; Sáez et al., 2016), (d) humid phases based on rodent middens ~24-26 °S (Díaz et al., 2012; Maldonado et al., 2005), (e) pre-CAPE and CAPE events ~20-24 °S (de Porras et al., 2017; Gayo et al., 2012; Pfeiffer et al., 2018; Quade et al., 2008), (f) alluvial fan active phases from other studies ~20-25 °S** *(Bartz et al., 2020a, 2020b; Vargas et al., 2006; Vásquez et al., 2018; Walk et al., 2023)***, (g) MAM ages (circles) and CAM ages (squares) within alluvial deposits this study ~25 °S; different colours represent the five different profiles (cf. Fig. 3-6), for sample P4-1 the mode is shown, MAM and CAM ages of P5-6 are slightly transparent since their associated process is ambiguous.**

The second and most recent phase of alluvial activity, recorded in profiles **P2**, **P3**, and **P5**, corresponds to an interval of both intensified and highly variable El Niño activity (cf. Fig. 9) starting at ~5 ka, which likely promoted an increased frequency of extreme rainfall events (Rein et al., 2005, 2004).







**Figure 9: Comparison of the luminescence ages (top) of all five profiles during the last 15 ka to independent proxies (bottom). MAM and CAM ages of geomorphic active surfaces are depicted as circles and squares respectively. Ages of geomorphic stable surfaces are depicted as triangles, with the MAM age for P1-2 and the mode for P4-2. The lithic flux rate, as an indicator for El Niño activity (Rein et al., 2005, 2004), and wet and dry phases, based on the *A. cinerea* pellet diameters from paleo-middens in the Atacama Desert (González-Pinilla et al., 2021), are provided as well.**

Besides these two main phases, additional evidence for alluvial activity is found in **P1**, **P2**, and **P5**. In **P1**, the ages of samples **P1-3** and **P1-4** are interpreted to reflect secondary depositional activity on top of the oldest alluvial fan generation at the onset of MIS 2. These findings contradict the findings of Walk et al (2023), who suggested a MIS 5 depositional age for the same fan generation based on cosmogenic nuclide dating of surface clasts. This discrepancy may on the one hand be explained by local reworking and surficial sediment redistribution through low-intensity sheet-wash. On the other hand, surface clasts in transport-limited cascading systems, in which short episodes of geomorphic activity are separated by long phases of geomorphic stability, are known to often contain inherited cosmogenic nuclide concentrations (Anderson et al., 1996; Walk et al., 2023). Based on our luminescence ages of the oldest fan generation, we consider an overestimation of this fan generation by Walk et al. (2023) likely. Although, the reasons for this discrepancy remain ambiguous, our findings underline the need of using complementing chronometers (e.g. TCN and feldspar luminescence) to refine the chronological architecture of such





alluvial deposits, and, ultimately, to detect possible dating biases. This is particularly true for complex geomorphic settings, where different dating techniques are affected by different types of errors (i.e. inheritance, heterogeneous bleaching).

Further contemporary depositional activity at the onset of MIS 2, as found for **P1**, was not observed elsewhere along the transect. Local climatic differences between the coastal plain and the Coastal Cordillera during this period are difficult to constrain based on our data. However, localised runoff on coastal alluvial fans may occur independently of regional climate
variability and is not necessarily related to alluvial activity within the Coastal Cordillera. Other studies reported alluvial deposition (Bartz et al., 2020a), activity in alluvial fans (Bartz et al., 2020b) and in alluvial terraces (Vargas et al., 2006) along the coastal zone during this interval (Fig. 8f). In addition, pollen analysis from rodent middens close to our study site indicate the occurrence of plants, and hence elevated humidity, in the Coastal Cordillera (Díaz et al., 2012) and at higher elevations of 2670 m a.s.l (Maldonado et al., 2005) during the same period (Fig. 8d). The aridity index shows a trend towards more humid
conditions during this phase, following an extremely arid period recorded toward the end of MIS 3 (Stuut and Lamy, 2004; Fig. 8a). Together, this suggests that the localised signals in **P1** may reflect broader climatic trends throughout the entire Coastal Cordillera, although Coastal Cordillera profiles studied here lack evidence of activity during this period.

In sample **P2-4**, two distinct phases of alluvial activity were identified. The younger phase corresponds to an interval of enhanced El Niño variability starting at ~5 ka. The older phase aligns with an earlier period of intensified El Niño activity
beginning ~13 ka (Rein et al., 2005, 2004; Fig. 9). Both phases also coincide with wet conditions inferred by González-Pinilla et al. (2021; Fig. 9) and with the formation of fluvial terraces (Gayo et al., 2023; Nester et al., 2007; Fig. 8b). The younger phase is further supported by evidence of alluvial deposition both along the coast (Vargas et al., 2006; Fig. 8f) and within the Cordillera (Vásquez I. et al., 2018; Fig. 8f). The older phase corresponds to debris-flow activity at a coastal alluvial fan (Bartz et al., 2020a; Fig. 8f). Additional indicators for the older phase include the Central Andean Pluvial Event (CAPE) II and pre-
CAPE events (de Porras et al., 2017; Gayo et al., 2012; Pfeiffer et al., 2018; Quade et al., 2008; Fig. 8e), regional groundwater discharge (Rech et al., 2002; Sáez et al., 2016; Fig. 8c), and pollen records from rodent middens in the Cordillera (Maldonado et al., 2005; Fig. 8d). These observations suggest that both phases of alluvial activity were likely driven by increased frequencies of extreme rainfall events linked to shifts in ENSO dynamics, highlighting the strong influence of climatic variability on sediment fluxes in the region.

The broad and multimodal $D_e$ distribution of sample **P5-6** and the granulometric and geochemical characteristics do not allow for identifying particular processes such as alluvial depositional activity or post-depositional mixing. However, its MAM age of $17.8 \pm 3.1$ ka indicates that bleaching of sediments in this unit ceased during the comparatively humid late MIS 2 (Stuut and Lamy, 2004; Fig. 8a). The CAM age of $48.9 \pm 8.3$ ka is consistent with the older depositional phases observed in profiles **P3** and **P4**, both situated above the marine boundary layer. Thus, the CAM age may reflect depositional activity under
regionally wetter climatic conditions during MIS 3. Between the CAM and MAM ages, multiple phases of increased humidity are evident in the aridity index (Stuut and Lamy, 2004; Fig. 8a), and are corroborated by all other regional proxies analysed in this study (Fig. 8b–f). Nonetheless, the processes associated with this $D_e$ distribution and the MAM and CAM ages remains ambiguous.





Against the background of these results, past and present alluvial depositional activity along the here presented climatic transect

appears to be primarily controlled by site-specific conditions. These include whether the sampling site is located within a geomorphologically active (**P2**, **P3**, and **P5**) or geomorphologically stable (**P1**, **P4**) segment of the respective alluvial deposit. The transect spans from the coastal plain (**P1**) through the fog-influenced zone of the Coastal Cordillera (**P2**) to the upper part of the Coastal Cordillera (**P3-P5**), where fog is virtually absent and rainfall frequency extremely low. Despite this climatic gradient, we observe no systematic spatial pattern in our observed geomorphic activity related to the geographic position of a

profile within the transect. This is consistent with the characteristics of transport-limited alluvial systems, where sediment supply exceeds the transport capacity of the system (Bovis and Jakob, 1999). In such settings, local controls such as slope morphology, channel connectivity, or sediment availability play a more significant role in determining depositional timing and surface reworking than external drivers alone. Consequently, deposition or alluvial reworking can occur within a spatially confined area without necessarily affecting downstream locations within the same catchment.

However, the present climatic gradient along the Paposo transect is indeed reflected by post-depositional processes in **P1**, **P2** and **P4**. In the lower elevations of the transect (**P1**, **P2**), post-depositional processes are likely related to bioturbation, while at higher elevations above the marine boundary layer (**P4**) they are presumably related to desert pavement formation. The probably bioturbation-related most recent phase of post-depositional mixing in **P1** falls within a wetter climatic interval characterised by comparatively strong El Niño (González-Pinilla et al., 2021; Rein et al., 2005, 2004; Fig. 9). Evidence for

depositional activity in profiles **P3** and **P5**, as well as phases of alluvial fan accumulation in more northern parts of the Coastal Cordillera (Vargas et al., 2006; Fig. 8f) support this interpretation. In contrast, post-depositional processes plausibly related to the (re)development of a desert pavement (Ugalde et al., 2020) in **P4** coincide with a phase of prevailing hyperarid conditions between ~8 and ~5 ka and weak and relatively stable El Niño activity (González-Pinilla et al., 2021; Rein et al., 2005, 2004; Fig. 9). Moreover, de Haas et al. (2014) emphasised that desert pavements cannot form under wet conditions

which lead to intensified weathering and surface reworking by runoff events. The absence of deposition during the last ~5 ka in **P1** and **P4** likely reflects the presence of younger alluvial fan segments. Surface runoff and associated sediment transport were redirected toward these adjacent, more active younger sections, resulting in the bypassing of older depositional surfaces. Consequently, **P1** and **P4** lack evidence of late Holocene fluvial activity, despite supposedly continued sedimentary dynamics within their fan systems.

Overall, our results support the growing body of evidence that, even under persistent hyperarid conditions, episodic increases in moisture availability were sufficient to drive localised alluvial processes. These findings emphasise the importance of integrating geomorphic, geochemical, granulometric, and chronometric data at high spatial resolution to understand landscape dynamics in extreme hyperarid environments. The high spatial resolution is of particular importance where geomorphological active and stable surfaces occur in direct vicinity. The identified post-depositional processes highlight that geomorphological

stable surfaces can retain climatic and geomorphic signals valuable for paleoenvironmental reconstruction. Moreover, careful interpretation of single-grain $D_e$ distributions can be used as a paleoclimatic proxy for the late Pleistocene and Holocene.



## 6 Conclusions

The combined analysis of luminescence dating, stratigraphy, geochemistry, granulometry and field observations of five profiles along a climatic transect in the Atacama Desert shows a complex interplay between depositional and post-depositional
processes in Atacama Desert soils and surfaces.

Although heterogeneous bleaching is a ubiquitous feature in alluvial deposits even near the surface, the high proportion of well-bleached grains at the surface allows two conclusions to be drawn: (i) a good bleachability of our samples, if grains are at the immediate surface, and (ii) a shallow effective bleaching depth (<2 cm). The different shapes of the single-grain $D_e$ distributions underscore the importance of interpreting luminescence ages in the context of surface stability and disturbance
regimes, rather than simple depositional ages for the alluvial fans. Our data further emphasise that single-grain dating is capable of distinguishing between depositional and post-depositional processes, which is not possible with multi-grain dating or TCN dating, due to their lower spatial and temporal resolution.

Along the investigated climatic transect, stratigraphic patterns are primarily governed by local geomorphic controls rather than regional climatic patterns. This results in a spatially heterogeneous surface evolution and varying degrees of soil and surface
activity and stability. Geomorphological active profiles (**P2**, **P3**, **P5**) show recent sediment accumulation and signs of vertical grain transport (**P2** only), while geomorphological stable profiles (**P1**, **P4**) preserve evidence of (soil) reworking superimposed upon longer-term landscape stability. Despite contrasting environmental settings, post-depositional mixing in **P1** and **P4** remains shallow, suggesting that soil processes in the Atacama are depth-limited regardless of the presence of vegetation.

The overall timing of soil and surface activity and stability aligns with a compilation of regional climatic reconstructions from
other studies. Especially the relatively strong and variable El Niño episode during the last ~5 ka and a relatively humid phase around 50 ka are recorded throughout the entire transect. This highlights the importance of integrating local-scale sediment dynamics and post-depositional processes when interpreting luminescence data for paleoenvironmental reconstructions in arid landscapes. Moreover, it demonstrates the potential of single-grain luminescence dating to resolve soil and surface processes that can be utilised as a high-resolution climatic proxies.

**Data availability**
Partly processed data will be made available on Zenodo.

**Author contribution**
LM: conceptualization, writing - original draft, data curation, formal analysis, investigation, visualisation

SM: conceptualization, writing - original draft, investigation, visualisation

SR: writing – review & editing, investigation, supervision

WM: writing – review & editing, formal analysis

SO: writing – review & editing, investigation



AP: writing – review & editing, investigation

TR: conceptualization, writing – review & editing, supervision, funding acquisition

**Competing interests**

The authors declare that they have no known competing financial interests or personal relationships that could have appeared to influence the work reported in this paper.

**Acknowledgements**

This project is affiliated to the Collaborative Research Centre (CRC) 1211 "Earth – Evolution at the Dry Limit" (Grant-No.: 575 268236062) funded by the German Research Foundation (Deutsche Forschungsgemeinschaft, DFG), Germany. We would like to thank Andreas Peffeköver his assistance during fieldwork. We further thank Sarah Spengeler, Johanna Steiner, and Johanna Schreiber for their contributions to the laboratory analyses. LM is also grateful to Eduardo Campos as well as Bárbara Blanco Arrué and her team for their kind help when we stranded along Ruta 5 without accommodation.

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
