# Peer review of "Geomorphological activity and stability of surfaces and soils formed in hyperarid alluvial deposits (Atacama Desert, Chile)"

_EGUsphere, 2025_

## Referee Comment (RC1)

Overview:

The manuscript entitled '**Geomorphological activity and stability of surfaces and soils formed in hyperarid alluvial deposits (Atacama Desert, Chile)**' submitted to ESD (EGU) by Linda Andrea Elisabeth Maßon, Simon Matthias May, Svenja Riedesel, Willem Marijn van der Meij, Stephan Opitz, Andreas Peffeköver, and Tony Reimann whether landform stability and depositional/reworking processes in landforms within a hyperarid setting (alluvial fans and ephemeral channels) can be studied by means of single-grain feldspar luminescence, geochemistry, and granulometry. The authors also claim that field observations are key to this aim; however, there is a need for a much sharp linking of basic sedimentological observations with the proxies used, which I find to be the major shortcoming of the manuscript in its present form.

This manuscript contains a tremendous amount of laboratory work on luminescence analysis; however, most of the final interpretations (processes and potential triggers) are still addressed only vaguely. This likely arises from the lack of basic stratigraphic descriptions (¿classic? sedimentological descriptions), which are presented to some extent in the Supplementary Material but are not convincingly connected to the proxies presented. As a consequence, the discussion is structured on a mix of ideas/concepts based on profile chronology and paleoclimatic interpretations that are not sufficiently justified by basic stratigraphic work. This issue alone should not, in my view, be sufficient to reject a manuscript; however, it is a major flaw that must be addressed. Unless greater attention and detail are devoted to the sedimentary characteristics of the selected profiles, any interpretation can be refuted and remains vague.

The authors have also chosen to frame all their working hypotheses under the assumption of a climatic gradient within a catchment located in the Atacama Desert Coastal Cordillera. Without additional evidence (including references) and without basic information on, for example, catchment lithology, fault activity, and Late Quaternary vegetation history, most of the claims made in the manuscript remain too speculative.

The selection of study sites (alluvial fans and ephemeral channels) needs to be clearly explained from the outset. Rather than searching for abiotic, biotic, and climatic gradients, I suggest simplifying the hypothesis towards evaluating long-term stability of alluvial fan surfaces in a landforms subject to different weathering histories, i.e., coastal fan) against tributary-junction fans located >1000 m a.s.l. without influence of i.e. marine aerosols. Additionally, when considering ephemeral channels (and therefore active – see references for events impacting the Atacama Desert in modern times: Wilcox et al. (2016), Scott et al. (2017), Cabré et al. (2022), among others I strongly recommend the lecture of basic literature on how such landforms form and respond to runoff events (see, for example, the books by Bull & Kirkby, 2000, and Graf (1988) on dryland rivers).

There is also a substantial body of literature on fog in the Atacama Desert and on biotic and atmospheric factors (such as fog and ENSO) feedbacks with landforms, including research by del Río et al. (2024, and references therein). Thus, I recommend a thorough review of this literature (maybe authors can point out others) to identify appropriate references relevant to the regional climatic setting in this Coastal region of the Atacama Desert. Although this region lies within the Atacama Desert, it exhibits very particular atmospheric conditions that need to be addressed accordingly.

I also recommend that the authors clearly identify the "paleoclimatic" proxies discussed (cited) in the paleoclimatic interpretation section by plotting their geographical locations. Doing so will show that most of the archives considered do not belong to the Coastal Cordillera physiographic domain, and therefore their applicability/relevance/influence should, at the very least, be explicitly discussed.

The conceptual understanding of alluvial fan formation presented in the manuscript lacks adequate referencing and is largely (almost exclusively) based on Blair and McPherson (2009) work. Given that the authors are working on a coastal alluvial fan, fan formation in relation to sea level, coastal uplift, and associated boundary conditions should be explicitly addressed by outlining the basic controls on fan development in such environments. Relevant works by Adrian Harvey (Harvey, 1999, 2002, 2018) or others (Hartley et al., 2005; Viseras et al., 2003; among others) could be consulted to strengthen this section. In this

regard, Figure 2 also requires refinement and updating using recent literature focused on the same region. Authors can check for details in the line-by-line section of comments.

The discussion of previous work done by Walk et al. (2020, 2023) on the same(similar) alluvial fan(s) is very limited and mainly restricted to surface dating results. I would have expected a more detailed comparison of the geomorphic processes identified by both Walk and the present authors in order to provide a more comprehensive understanding of what geomorphic stability means in hyperarid environments and in particular in coastal alluvial fans. The authors could consider recent work (Walk, 2025), which offers fresh insights into coastal Atacama Desert alluvial fan systems.

Before proceeding to line-by-line comments, I would like to draw the authors' attention to the misuse or incorrect use of the term lithological unit when referring to stratigraphic layers, beds, or horizons in their profiles. The term lithological unit should be restricted to [appropriate stratigraphic usage], and its current application throughout the manuscript is misleading. You can check (as an example) the definitions provided by the international commission on stratigraphy (link).

It is also unclear how the XRD-derived geochemical data are ultimately integrated into the discussion of the results (apart from the PCA plot). It is concerning that the authors state in the discussion that the site-specific geochemical results are not conclusive and do not discuss potential limitations (at least). This raises the question of whether there was a clear rationale for applying XRD analyses in the first place.

Specific comments & Minor comments:

L9: *"Hyperarid earth surface processes"* as a keyword is not reflected in the manuscript content. This is not a review paper; therefore, I recommend restricting the keywords to processes that can actually be addressed and constrained by the methods applied in this study.

L12: Please clarify what is meant by *"exceptionally high geomorphic stability"* and specify which parameters or metrics this assessment is based on.

L14: This statement is not entirely accurate. Muñoz-Farías et al. (2023) and also Binnie et al. (2018) demonstrate that surfaces commonly referred to as "stable" can record markedly different sedimentary and sediment-capture histories.

L16: Please explain how the proposed "wetter" Quaternary episodes are reconciled with the episodic sedimentation events recorded by debris flows in the same coastal alluvial fans studied here (see Bartz et al., 2020; Walk et al., 2020).

L18: Are the authors truly determining the drivers of geomorphic activity? The manuscript does not present a clear end-member or comparative framework that would support this claim.

L20: Do the authors intend to apply the 2 cm bleaching-depth rule uniformly across the entire Atacama Desert (see statement at L385: *"we can infer that the bleaching depth …"*)?

L21: Profiles—not landforms—are grouped as active or stable based on the multi-method approach. This leads to a more general concern: although the manuscript repeatedly refers to a multi-method approach (and this is evident in the Methods section), I still miss a clear linkage between the different methods and how they jointly support or constrain the interpretations presented. There are clues -presented in figure S6 by the authors- to how to proceed.

L23: Please clarify which vertical grain-transport mechanism is being considered.

L24: Aren't desert pavements problematic for OSL dating? Please check and cite relevant references. Recent work of Fuchs et al. (2025) might give the literature search but is not exclusive. See: Carr et al. (2024), Fuchs et al. (2015),  among others.

L24: This issue may arise because the selected catchment does not exhibit a clear climatic gradient.

L25–28: There are not enough arguments provided to support this statement. I will return to this point in the Discussion section.

L31–32: The study area does not fall within the extremely hyperarid core that is claimed. Please check and revise this statement using appropriate references.

L35–44: This paragraph needs to be better structured and should provide a concise framework for the studied region, rather than a broad introduction to paleoclimatic studies in the Atacama Desert that may have little or no relevance to this study. Please also check the spatial extent of areas influenced by increased moisture; the Altiplano should probably not be included here.

L45–53: The Introduction reads as mixed and unfocused. Again, the climatic transect concept does not work convincingly. Please clarify that the study focuses on alluvial deposits from different landforms with well-established depositional and reworking processes.

L58: The references by Zinelabedin et al. (2022, 2025) focus on dating calcium sulphate wedges in distal parts of large alluvial fans in the Central or Intermediate Depression. Apart from the use of single-grain luminescence dating, their relevance to the present study is unclear. References such as del Río et al. (2018) and Bartz et al. (2020) may be more appropriate for the stated aims. The second aim of the study also remains questionable. Is there a clear and unambiguous application of the combined techniques that directly addresses this aim? The stated aims need to be revisited.

L60–62: This objective cannot be achieved without detailed stratigraphic field descriptions.

L64: Figure 1: On what criteria are the biotic–abiotic domains defined? Are there references that support this subdivision?

L70: Providing general regional insights into hyperaridity is only necessary if the study aims to investigate processes affecting landforms in hyperarid settings. This does not appear to be the case, as the study area is not located within the hyperarid core of the Atacama Desert. Please support this with references, including maps based i.e., on pluviometry records.

L74–80: I strongly recommend providing more detailed information on fog influence in the Coastal Cordillera and its effects on landforms. The proposed coastal–inland subdivision within the catchment could be strengthened by incorporating references from del Río et al. provided above.

L80: Please explain or specify what is meant by "on a local level".

L86: The climatic data are acceptable, but how relevant is seasonal moisture variability in the context you consider? How does this relate to actual precipitation events? A more in-depth understanding of the atmospheric processes operating in the selected study area is required.

L90: What do you consider an "extensive drainage basin"? Is the reference to Walk et al. (2023) applied to the catchment or to the alluvial fan?

L91: What do you mean by a "heterogeneous lithological composition"? For example, are you referring to a conglomerate with a polyphasic clast composition? This needs to be clarified, and more detailed information on catchment lithology should be provided.

L91–92: Please clarify whether the broad, shallow valleys are filled with conglomerates or whether these valleys incise Neogene intramontane basins.

L93: Is there an available reference for the *Matancilla* intrusive complex?

L96: Please check for a typo in the citation.

L97: What is meant by a "fault that crosses the area"? Does this description make geological sense in this context? You also need to clarify whether you are referring to the Atacama Fault System (see relevant references) or using informal terminology. If the Atacama Fault System is considered, there is no need to describe individual minor faults, as the AFS is a composite fault system rather than a single fault trace. A more thorough work/presentation of the geology of the study area is required.

L100: What do you mean by "except for P2"? How do you know that faults in this region are active? Please provide references or field evidence for recent activity or ruptures.

L101: What controls the elevation profile of the transect? To what physiographic or geological characteristics is the reported ~110 m elevation difference related?

L105: "Geomorphological survey" is a very broad term. Please specify more precisely which aspects were investigated. What is meant by "site documentation," and what does this include?

L106: Are you describing soil profiles or sedimentary logs? A combination of both seems necessary to address the main research question. Figure S6 shows particularly promising material; rather than leaving it as supplementary information, you may want to reconsider using such figures as key elements of the manuscript. You also state that profiles were excavated along a climatic transect, yet earlier you emphasize hyperarid climatic conditions. Does this imply the presence of a climatic gradient? Please justify this statement and provide supporting references.

L107: I do not understand the sentence beginning with "All profiles ...". Please rewrite it for clarity.

L110: How much material was recovered from each sampled layer? Is it possible to provide an average amount? This is not a critical concern, but it would be useful for reproducibility.

L114: Figure 2: What is meant by an "uplifted mountain block"? Is the coastal cliff interpreted as a fault trace? If so, is there evidence supporting this observation? Why are sub-catchments delineated? Is there a specific geomorphological relevance—such as catchment piracy or other constraints—that justifies this subdivision? Could potential drainage capture have influenced the evolution of the landforms whose development you aim to reconstruct? Please refer to works dealing with that geomorphological topic for insights into this concept, which has not been considered in your investigation. Again, on what basis is the biotic–abiotic transition defined?. More generally regarding Figure 2, I recommend using the original source of the figure rather than a previously modified version. Please clarify whether the figure represents the general structure of an alluvial fan or is intended to depict your specific study area. Please also clarify what is meant by "the left segment." Do you mean the northern or southern part of the fan, or another directional reference? Additionally, is there an explanation for the occurrence of sheetflood deposits on the left and debris-flow deposits on the right? From a sedimentological perspective, this is a pattern that requires further explanation.

L120: Figure 3 caption: The statement "Profile units apply only to the respective profile" is unclear. Please rewrite it for clarity. The term "lithological units" is incorrectly used here and should be replaced with "stratigraphic layers."

L126: How do the geochemical data help in deciphering depositional processes? Please clarify. This also raises a more general concern regarding the use of geochemistry: what is the underlying purpose of these analyses? Why were sedimentary facies not linked to end-member definitions using the applied proxies? Stating that geochemistry is indicative of depositional processes is either incomplete or, as currently presented, incorrect.

L128: When you refer to "insufficient material," is this something that could have been anticipated, or are there specific indicators of how much sediment would have been required? Fieldwork limitations are understandable, but simply stating "insufficient material" does not clearly convey whether this is a sampling limitation or a methodological constraint.

L139: The wording "attention was ..." is inappropriate here. Please rewrite the sentence for clarity.

L140: Is there any indication of how representative the selected grain-size fraction is for the different deposits studied (for luminescence purposes, of course)?

L156: Given that you rely heavily on De distribution values, please consider expanding on how these distributions relate to the sedimentary facies present in the studied profiles. Again, your figure S6 is very promising in that regard.

L172: Question: Do you consider these samples to be "modern analogue" samples or not? Please clarify.

L176: Were De values close to zero or negative excluded from the analysis?

L178: What is meant by "upward transport"? Please clarify this term.

L182: Could this observation be related to the development of desert varnish on desert pavement surfaces? If so, you may want to provide a reference or explore this as a possible explanation for incomplete bleaching.

L192: What do you mean by an "exemplary grain-size distribution"? Please clarify.

L195: Is the "mean dust frequency distribution" an indirect way of suggesting that the fine sediments accumulated in the profiles are allochthonous? The relevance of dust-trap sediments from previous studies is not clear to me in the context of this manuscript. Perhaps you could expand on potential extra-catchment sediment sources and their relevance to your interpretations? Although the grain-size fraction used for luminescence differs from that of the dust samples, dust input may still strongly influence the geochemical results. I have not been able to determine whether the geochemical analyses considered bulk sediment samples or a particular grain-size fraction. If so, this should be stated more explicitly in the methods section.

L196: The manuscript then shifts to weathering indices without clearly linking them to the stated research questions. Immediately afterward, it is stated that these indices are not well suited for desert environments, which makes their inclusion appear contradictory and confusing. Chemical weathering topics require a clearer framework or justification in this context.

L203: Figure 5b: I would avoid comparing the evolution of profiles from alluvial fans and ephemeral channels together. P1 clearly reflects the effects of a stable surface with negligible surface erosion; however, what needs to be explained is how P4 compares to this profile. This comparison could then be used to strengthen the hypothesis of a persistent long-term climatic gradient. You also state at L200 that comparisons between profiles are not possible due to differences in parent material, presumably reflecting lithological variability within the catchment. This again highlights the need to better characterize catchment lithology, as mentioned in earlier comments. If lithology exerts a strong control on the observed differences, the climatic interpretation becomes less robust, and this issue therefore needs to be addressed more thoroughly.

L209: Again, is the granulometry entirely a product of weathering or differences in regolith from different rock units, or is there a component derived from external sources? This possibility should at least be considered, even if there are no data to support one explanation over another.

L218: When you refer to "sufficiently bleached," is this your own assessment, or are you using a reference value? If it is the latter, please provide appropriate references. The same concern applies to the term "effectively bleached" at L223.

L235: You report negative De values within the figure. Is there any explanation for this? What is meant by the "dust-trapping phase"? Please clarify. Why are dashed lines drawn between points in the figure? Is there any evidence of continuous sedimentation in these landforms that could be discussed?

L241: When you refer to the "oldest" material, relative to what is this defined? Is there a potential preservation bias that should be considered?

L248: Instead of stating "Unit I is the only one extensively penetrated...," I recommend rewriting to something like "Unit I shows evidence of bioturbation by roots" or a similar description that specifies the observation.

L249: Are the features you describe modern roots, or are them root moulds, tubules? Please clarify.

L251: Can the material exchange you mention be attributed to a particular feature identified in the field? For example, is there evidence of desert pavement formation?

L255: To define depositional phases, it is necessary to clearly identify the sedimentary processes present. Without this, the interpretation risks being inconsistent.

L259: Is a deeper discussion warranted regarding the selection of one model over another for your interpretations?

L260: The definition of aggradation phases remains complex without additional stratigraphic information.

L261: Can you list the initial soil formation processes observed in your profiles?

L264: This comparison with Haug's work is not valid because their study focuses on fans in the Intermediate Depression, which are not comparable to coastal fans. You also state that the oldest surface might have been active—does this imply it is no longer the oldest surface? Please clarify.

L266: When discussing inherited ages, could you also provide insight into the potential reworking of Neogene conglomerate units and how this may influence your interpretations?

General comment on the last paragraphs: I think a much more detailed study of the observed sedimentary facies and/or soil formation is needed in order to interpret the ages in the most 'honest'¿? and meaningful way. The depositional phases you mention, based solely on ages without stratigraphic context, are somewhat arbitrary and are not clearly supported by the chosen proxies. Perhaps the available data could have been used to develop a sedimentary end-member model that would allow you to discriminate between different sedimentary processes responsible for the ages. In other words, do you understand what the ages you present actually represent? Do you know if they correspond to a sedimentation event followed by reworking, or if they represent a single deposit that has been preserved without reworking to the present day? Much more work and description regarding the sedimentology is required. While the OSL laboratory work is impressive, its significance is best appreciated when supported by thorough sedimentological context.

L273: Can you expand on the different De peaks and the sedimentary characteristics of the sampled layers?

L276: Can you provide references supporting recent activity of this fault? Also, which sedimentary processes would be responsible for this mixing? Please clarify.

L277: Does this "marine boundary" remain stable throughout the Pleistocene and Holocene? Can you provide references to support this?

L283: Please clarify and explain exactly what aligns with the lithology.

L287: A general comment on the previous lines: once you have defined sedimentary facies for the locations sampled for luminescence, you can relate how De peaks correspond to each sedimentary end-member (facies). This could itself be a meaningful result and would provide stronger support for your interpretations (check your figure S6 as an example to do this).

L291: Incorrect use of the term "paleosurface." Are you referring to a hiatus surface in sedimentation? Also, can you clarify which roots you are referring to—actual roots? Do they influence all layers of the profile? Please clarify.

L308: Older relative to what?

L330: That is interesting. Perhaps you could frame some of your De results together with the sedimentary facies to which they correspond. Again, check Fig. S6 example.

General comment for the last lines: Are the mechanisms of trapping variable? Is there a potential increase in fines due to breakage on upper surfaces, exposing previously formed vuggy horizons characteristic of desert

pavements? Are you recording increased erosion or deposition as a result? This could be relevant depending on the landform. Both mechanisms are known to operate in alluvial fans under the same climatic and tectonic regimes, so what is your explanation?

L340: Is this a profile of an ephemeral channel? There is a clear need to provide a much more detailed sedimentary description. For example, in P5, Figure 3 shows open-framework gravel or couplets, which are known to be deposited by a single runoff event; this needs to be carefully checked.

I somehow already said that in the overview comment but: now in L340–354 that applies to the entire manuscript: Whenever you use the term "unit," this is simply a sedimentary layer, and I do not see the relevance of defining it as a unit. A "unit" of what? Early in the manuscript, you refer to lithological units, but this is clearly incorrect. By definition, a lithological unit should be mappable under lithostratigraphic criteria; your "unit" definitions do not meet this standard.

355: As I mentioned in the Methods section and earlier comments, I think each profile first needs a sedimentary interpretation (e.g., alluvial fan, channel, etc.). Then, comparing them solely based on biotic–abiotic domains introduce a bias toward expected outcomes from your dataset. This manuscript should start from the basics of stratigraphy before moving into complex interpretations of De distributions for each sampled layer. Without a more detailed sedimentary description, even the most impressive laboratory work cannot effectively address the scientific question of how landforms form, evolve, and are present today. By establishing the sedimentological framework first, you could more effectively integrate your De analysis and geochemistry with the sedimentary end-members or facies you define. A combination of the "theoretical" Figure 7 and/or your abanico plots (figures available within your supplementary materials), together with defined sedimentary facies, would be extremely valuable.

L385: This statement reads somewhat too ambitious for a local study in a particular catchment of the Coastal Cordillera with specific lithologies. My impression is that the 2 cm bleaching-depth value is unlikely to be universally applicable across the entire Atacama Desert.

L388: Regarding post-depositional mixing processes, and considering your earlier mention of seismic activity, would it make sense to integrate the effects of seismic shaking here? There may be references that could support this consideration.

L390: I am missing -again- more detailed explanations of the rock types present in your catchment.

L392: It reads odd, despite detailed fieldwork on these profiles, the conclusion is essentially "not possible." This is not meant as a harsh criticism, but I believe your observations could be explored in more depth to provide a more elaborate interpretation.

L396: This is puzzling because you provide a PCA where there is room to improve interpretation, yet you now state it does not convey much.

L399–402: After examining the landscape and sampling both active ephemeral channels and abandoned fan surfaces (i.e., no drainage developed on top), this alone could have led to the conclusions you present using a dense dataset. The question here is: should one rely on classical geomorphological mapping criteria, or conduct extensive analyses that do not appear fully conclusive, yet lead to a similar conclusion? Your approach and its benefits are not entirely clear.

L403–405: Proper stratigraphic descriptions are necessary to understand the sedimentary processes operating in these profiles. For example, recent events controlled by current base level (e.g., sea level) may not be recorded or preserved, meaning much of the Holocene could be missing. Consequently, any paleoclimatic interpretations would be incomplete. What are your considerations in this regard?

L407: Could your observations be simply explained by the fact that the landforms are simultaneously experiencing both erosion and deposition, reflecting a dynamic system?

L413: By "alluvial deposition," do you mean sediment transport?

L414–415: By "incision," do you mean channel incision?

L417: What "topographic barrier" are you referring to?

L418–419: Can you provide estimates from historical datasets? (longer than the ones you show in figure S1?)

L422: Perhaps, but your conclusions are not strongly supported by field observations. You may want to expand on this.

L427: Be cautious: a single runoff event can remove >7 m of channel infilling sediments (see examples in El Salado river (Wilcox et al., 2016; Lazo et al., 2025; Contreras et al. 2024) and close-by ephemeral channels close to your study area by Cabre et al., 2022. Therefore, the deposits you observe may not present the full picture because of the removal of large volumes of sediments.

L468: There is the publication by Walk (2025) on the same fans, which you may want to consider in your resubmission.

L477: How close are these features? Would it be useful to include them in a map?

L495: General comment from previous lines: from five profiles, for which you do not provide detailed information about the responsible sedimentary processes, you proceed to regional climate interpretations. This seems inappropriate.

L508: Previously, you stated that the area represents a climatic gradient. I have not found any supporting evidence for this yet.

L511: Aguilar et al. (2020) shown this for the Atacama Desert so you can use local references.

L513–514: Do you mean that the characteristics of rainfall are heterogeneous and, therefore, sediment transport is also variable? Consider checking recent events (Cabre et al., 2022, Atacama) or classic studies from Nagev Desert for reference of spatially stochastic distribution of rainfall in drylands.

L517: Has this boundary been formally defined? Desert pavements are observed in coastal fans (e.g., Salar Grande; Hartley et al., 1990).

L520: The writing here is unclear. Please explain.

L521: When you mention post-deposition, isn't reworking the simplest explanation? Small-magnitude events might not be recorded in stratigraphy but could still produce significant erosion of deposits.

L524: Desert pavements have indeed been described in the Coastal Atacama (Hartley et al., 1990).

L527–529: So, what is the interpretation? If there is bypassing, there may have been activity, but in your sampled locations deposition is not recorded. A better sampling strategy -or limiting the applicability of your results to paleoclimate interpretations- seems more reasonable.

L531–536: I would suggest revisiting this section in the final version. Try to ground your conclusions more firmly in your observations and be honest about the methodological limitations, which are already clearly presented.

Conclusions: The conclusions need to be revised after editing the manuscript to reflect the points above.

L579: I'm very sorry to read that.